# What Serum Sodium Concentration Is Suggestive for Underhydration in Geriatric Patients?

**DOI:** 10.3390/nu12020496

**Published:** 2020-02-15

**Authors:** Zyta Beata Wojszel

**Affiliations:** 1Department of Geriatrics, Medical University of Bialystok, Fabryczna str. 27, 15-471 Bialystok, Poland; wojszel@umb.edu.pl; Tel.: +48-85-869-4982; 2Department of Geriatrics, Hospital of the Ministry of Interior and Administration in Bialystok, Fabryczna str.27, 15-471 Bialystok, Poland

**Keywords:** electrolyte, osmolyte, screening, prediction of underhydration, calculated osmolarity, older people, receiver operating curve analysis, water intake

## Abstract

Dehydration is a concern among aging populations and can result in hospitalization and other adverse outcomes. There is a need to establish simple measures that can help in detecting low-intake dehydration (underhydration) in geriatric patients. The predictive performance of sodium, urea, glucose, and potassium to discriminate between patients with and without underhydration was evaluated using receiver-operating characteristic (ROC) curve analysis of data collected during the cross-sectional study of patients admitted to the geriatric ward. A total of 358 participants, for whom osmolarity could be calculated with the Khajuria and Krahn equation, were recruited to the study. Impending underhydration (osmolarity > 295 mmol/L) was diagnosed in 58.4% of cases. Serum sodium, urea, fasting glucose, and potassium (individual components of the equation) were significantly higher in dehydrated participants. The largest ROC area of 0.88 was obtained for sodium, and the value 140 mMol/L was found as the best cut-off value, with the highest sensitivity (0.80; 95% CI: 0.74–0.86) and specificity (0.83; 95% CI: 0.75–0.88) for prediction of underhydration. The ROC areas of urea, glucose, and potassium were significantly lower. Serum sodium equal to 140 mmol/L or higher appeared to be suggestive of impending underhydration in geriatric patients. This could be considered as the first-step screening procedure for detecting underhydration in older adults in general practice, especially when limited resources restrict the possibility of more in-depth biochemical assessments.

## 1. Introduction

Several factors can lead to impaired hydration in older patients. Age-related physiological changes make this population especially prone to dehydration. These include significant alterations in the ability to maintain fluid and electrolyte homeostasis, decline in fluid reserves, inability to concentrate urine due to changes in renal function, or reduced thirst and fluid intake relative to younger individuals [1,2,3]. Additionally, multimorbidity, polypharmacy, functional dependency, and insufficient caring situations strongly increase the risk of improper hydration in older people [4]. Recently, it has been recommended to use the term “underhydration” to determine the state of low-intake dehydration in older patients [5].

Untreated underhydration negatively affects health and wellbeing of the older individual [6]. It is also an important risk factor of mortality during periods of disease and warm weather [7,8]. Prevention, monitoring, and management are critical to preventing dehydration-associated problems, but this is more difficult in older people. Physical signs of dehydration in this population show poor sensitivity. In the case of patients aged 65 years or older, admitted to the teaching hospital in Japan due to an acute medical condition, the sensitivity of physical signs of dehydration—such as dry mouth; sunken eyes; decreased skin turgor; dry axilla; decreased consciousness level; and delayed capillary refill time—was equal to 56% (for dry axilla) or lower [9]. The diagnostic accuracy of commonly used signs and symptoms of dehydration (such as self-reported feelings of thirst and well-being; mouth, skin, and axillary dryness; skin turgor; sunken eyes; capillary refill; blood pressure upon resting and after standing; body temperature; and pulse rate) was also low in the study performed in care home residents aged ≥ 65 years in the United Kingdom [10]. Because of this, very often, those who are dehydrated are not being identified or the hydration status is misdiagnosed and neglected [11,12,13]. 

Both physiologically and clinically, water and sodium metabolism are closely related. A sodium concentration within the range of 135–145 mmol/ L is treated as normal. Although in the general population, combined sodium and water deficits are far more frequent than isolated deficits of either constituent, in patients who cannot respond to thirst by the voluntary ingestion of fluids, or in those with an inadequate thirst response (like in many geriatric patients), relatively pure water depletion (dehydration) leading to hypernatremia is observed [13,14]. Hypernatremia outcomes observed in older people are in parallel to those described for dehydration. These are associated with multimorbidity and increases in in-hospital and post-discharge mortality rates [15,16]. 

Inadequate fluid intake is often equated with the presence of hypernatremia, defined as a serum sodium concentration exceeding 145 mmol/ L. However, in clinical practice, cases of older patients with dehydration symptoms and without concomitant hypernatremia are often observed. Therefore, there is doubt as to whether only a serum sodium concentration exceeding 145 mmol/ L should be considered as abnormal and suggestive of insufficient hydration in older patients.

Sodium is the basic electrolyte that determines plasma osmolality and is one of the individual components of the equations used to assess plasma osmolarity, in addition to glucose, urea, or potassium. The calculated serum osmolarity (approximating serum osmolality-the best indicator of static dehydration) is now the first suggested step for the assessment for underhydration in older patients, and should be followed by a serum osmolality measurement for those identified as at high risk. Patients are considered to be at high risk when calculated serum osmolarity is above 295 mMol/L [10,17]. Although the implementation of such a procedure in hospitals or care institutions can be feasible and cost-effective, it could be more difficult in cases of community dwelling older people, being under the care of general practitioners. There is a need to establish simple measures that could help in the screening for low-intake dehydration in geriatric patients and alert carers early enough that fluid supply is insufficient and that there is a risk of underhydration. This would allow for timely intervention before low-intake dehydration occurs [18]. Additional questions that may be asked are whether other, selected osmolytes can also possibly be used for screening dehydration resulting from insufficient fluid intake; and what concentration values of these substances should be used as limit values for the initial assessment of underhydration in older patients.

Therefore, the aim of the study was to evaluate the usefulness of blood sodium, urea, glucose, and potassium measurements as single tests that can guide the detection of low-intake dehydration (underhydration) in geriatric patients, and to establish their best cut-offs that could be used for this purpose. 

## 2. Materials and Methods 

The study was based on the secondary analysis of data collected during a cross-sectional study on frailty and multimorbidity in geriatric ward patients. All cases of patients consecutively admitted to the Department of Geriatrics of the Hospital of the Ministry of Interior and Administration in Bialystok, Poland, with serum sodium, potassium, glucose, and urea values assessed on admission, and for whom serum osmolarity was able to be calculated, were included in the analysis [19,20]. 

Data collected from the participant, medical notes, and/or relatives included demographics, co-morbidities (of 15 chronic diseases: peripheral arterial disease, ischemic heart disease, myocardial infarction, chronic cardiac failure, hypertension, atrial fibrillation, stroke, chronic obstructive pulmonary disease, diabetes/ prediabetes, neoplasm, dementia, parkinsonism, chronic arthritis, chronic renal disease, osteoporosis), number of medications taken at admittance, history of hospitalizations and falls in the last 12 months, and functional assessment results conducted within the routine comprehensive geriatric assessment during the patient’s hospital stay (the Barthel activity of daily living index (ADL) score [21], the 6 instrumental ADL items of Duke OARS scale score (IADL) [22], cognitive function assessed using the Abbreviated Mental Test Score (AMTS) [23], emotional health assessed with Geriatric Depression Scale (GDS) [24], nutritional health assessed with the Mini Nutritional Assessment- Short Form (MNA-SF) [25], frailty status assessed with 7 item Canadian Study of Health and Aging Clinical Frailty Scale (CFS) [26]). Polypharmacy was defined as 5 or more drugs taken. Multimorbidity was defined as 5 or more diseases out of 15 listed (as above). 

Blood sampled at admission was used to measure serum concentrations of sodium, potassium, urea and glucose, creatinine, eGFR. Osmolarity was calculated with the Khajuria and Krahn equation for osmolarity assessment [osmolarity = 1.86 × (Na^+^ + K^+^) + 1.15 × glucose + urea + 14; each component measured in mmol/L] [27]. Underhydration was diagnosed if calculated osmolarity was above 295 mmol/ L, a threshold identifying most adults with low-intake dehydration [28]. 

### 2.1. Statistical Analysis

Data were collected and analyzed using IBM SPSS Version 18 Software suit (SPSS, Chicago, IL, USA) and STATISTICA 13.3 software package (TIBCO Software, Palo Alto, CA, USA), and presented as means and standard deviation for normally distributed variables, and as medians and interquartile range for not normally distributed continuous variables, and the number of cases and percentage for categorical variables. Shapiro Wilk test was used to assess the distribution of variables. Statistical significance of differences between underhydrated and euhydrated patients was determined using the *χ*2 test (for comparing proportions), independent samples *t*-test (for comparing means) and Mann–Whitney *U* test (for comparing medians). Statistical significance of differences between underhydrated and euhydrated patients was determined using the *χ*2 test, independent samples *t*-tests, and Mann–Whitney *U* tests. The predictive performance of individual components of the equation used for calculating the serum osmolarity was evaluated using receiver-operating characteristic (ROC) curve analysis. Area under the curve (AUC), sensitivity, specificity, positive and negative likelihood ratio, and the best cut-off to maximize sensitivity and specificity with two-sided 95% CIs were calculated. Differences between AUCs were assessed with the Z-test. A *p* value of less than 0.05 was regarded as significant. Missing values were omitted and statistics in such cases were calculated for the adequately reduced groups.

### 2.2. Ethics Approval

The source study was approved by the Ethics Committee at Medical University of Bialystok (no R-I-002/305/2013). All the procedures performed in the study were in accordance with the ethical standards of the Medical University of Bialystok research committee and with the Helsinki declaration and its later amendments. All study participants gave their informed consent to participate.

## 3. Results

A total of 358 (86.1%) participants were recruited to the study after screening 416 admissions. Reasons for non-inclusion included situations in which there were no data on serum sodium (15 cases), urea (42 cases), potassium (15 cases), or fasting glucose (21 cases) tested on admission. 

Participants’ median age was 82 years (IQR, 78–86 years), with 76.5% identifying as female, 96.9% community dwelling, and 30.3% reporting living alone. The median number of confirmed chronic conditions was 5 (IQR, 3–6) and the median number of medications taken was 7 (IQR, 5–9). 

### 3.1. Hydration Status

The mean value of counted osmolarity was 295.6 mmol/L (SD, 7.8 mmol/L) in the study group. In 209 (58.4%) participants, osmolarity was above 295 mmol/ L, therefore, according to the criterion applied, they presented with impending low-intake dehydration at admission. There were no significant differences in the age, gender, place of living, and living alone in the scores of Barthel Index, IADL, AMTS, GDS and MNA-SF (Table 1), although biochemical differences and differences in number of chronic diseases and medications taken were demonstrated between those euhydrated and those with impending underhydration at admission to the department. The median values of sodium, urea, and fasting glucose, the mean value of potassium, and the median values of chronic diseases and medications taken were significantly higher in dehydrated participants. Hyponatremia (serum sodium below 135 mmol/L) was observed mainly in patients with calculated osmolarity below 295 mOsm/L, and there were no cases of hypernatremia (serum sodium above 145 mmol/L) in the study group.

### 3.2. Predictive Discrimination of Underhydration with Serum Sodium, Urea, Glucose, and Potassium

An ROC curve analysis was performed to test the predictive discrimination of patients with and without underhydration with the individual components of the osmolarity equation (Figure 1 and Table 2). The largest ROC area of 0.88 was obtained for sodium, and the value 140 mmol/L was found as the best cut-off value for this component of the equation; it yielded the best combination of sensitivity (0.80; 95% CI: 0.74–0.86) and specificity (0.83; 95% CI: 0.75–0.88) for prediction of impending low-intake dehydration. The next largest ROC area of 0.72 was obtained for urea, but it differed significantly from that for the sodium (difference between areas—0.154; CI: 0.100–0.208, *p* < 0.001), and sensitivity of urea was noticeably lower (0.75; 95% CI: 0.68–0.80). The ROC areas of glucose and potassium were similar and significantly lower than those for sodium and urea.

## 4. Discussion

The study demonstrated that, when using calculated serum osmolarity as a marker of hydration, impending underhydration was present in more than half of older adults admitted to the geriatric department; up to 58.4% of those admitted to the ward of geriatrics had osmolarity greater than 295 mMol/L. These results are consistent with other authors’ observations, although numbers of confirmed dehydration (diagnosed with measured serum osmolality) would certainly have been lower. For example, in the HOOP prospective cohort study, 37% of older patients acutely admitted to a large UK teaching hospital were diagnosed with hyperosmolar dehydration (defined as measured serum osmolality greater than 300 mOsmol/kg) [8]. 

Impending underhydration was observed mainly in patients with multimorbidity and polytherapy (see Table 1). In this respect, the completed study confirmed the results of other research, that people burdened with morbidity and those who take various medications, were more threatened with dehydration [29]. Contrary to other authors’ results, no significant association of underhydration with functional—mental and physical—disability was observed. One of the reasons can be the generally high prevalence of frailty and co-occurrence of different geriatric problems in this group as a whole [20]. The study population was not randomly sampled from the general population of community-dwelling older people, but it was a convenient sample of frail older patients referred to hospital—predominantly by general practitioners—for comprehensive geriatric assessment to be performed. The minority of participants were admitted because of a sudden, severe deterioration in general health. This may influence the results to a large extent.

Hospital admissions in older patients for underhydration, frequently resulting in adverse outcomes, may reflect the poor quality of care provided in community settings [30]. Therefore, given the high incidence of insufficient hydration in this population, a clear need exists for an improvement in underhydration detection and diagnosis in the primary care [20]. It would be also necessary to make general practitioners, who look after older patients burdened with multimorbidity and disability on a daily basis, more sensitive to this problem. The diagnosis of underhydration is often delayed, because a high percentage of older patients—in a situation where dehydration can be confirmed based on objective diagnostic tests—have no attributable symptoms of improper hydration [10,31].

Although osmolarity calculated with the Khajuria and Krahn equation has been recommended in the recent ESPEN guidelines as the first step in the assessment of the low-intake dehydration in older patients, its usefulness as a screening test may be limited in everyday clinical practice carried out outside hospitals and care institutions. If it were to be effective, doctors should order older patients to perform sodium, urea, glucose, and potassium blood tests on a regular basis. There is a lack of appropriate organizational solutions enabling the automatic calculation of osmolarity in laboratories cooperating with GPs, and it can hardly be expected that family doctors, often working within time constraints, would frequently do such calculations on their own. There is also a tendency to limit expenditures on diagnostic tests, if they are not perceived as really necessary. It would be reasonable to propose a simpler approach, based on the frequently performed biochemical test, that can be used to screen patients at risk of dehydration in family practice, and trigger the attention of geriatric patients’ carers.

Water and sodium homeostasis are closely related, and an insufficient supply of free water is associated with increasing serum sodium concentration in older patients. In the study by Shimizu et al., assessing the correlation between dehydration—defined as calculated osmolarity above 295 mOsm/ L—and several signs and symptoms of dehydration, from blood tests, only the measured osmolality and the mean concentration of serum sodium were significantly higher in the dehydrated group than in the non-dehydrated group [9]. Serum sodium above the ‘normal range’ (135–145 mmol/L) is well-known to indicate dehydration, but contrary to expectations—although impending underhydration diagnosed with calculated osmolarity was found in more than half of the elderly admitted to the geriatrics ward—no cases of hypernatremia were observed in the current study. However, the results confirmed that serum sodium and—to a lesser extent—serum urea could be used as single tests pointing to impending underhydration. The study confirmed that even lower serum sodium should worry us. If serum sodium concentration was equal to 140 mmol/ L or higher, or if serum urea concentration was equal to 7.5 mmol/L (45 mg/ dL) or higher, impending underhydration was able to be predicted, with sensitivity of 80% and 75%, respectively, although the ROC area was significantly greater for serum sodium measurement. In addition, specificity of sodium as a predictive factor was noticeably higher (83% versus 72% of urea). This could support the suggestion that serum sodium might be treated as the main individual predictive factor for impending underhydration in older patients. This seems understandable considering that sodium accounts for approximately 95% of the osmotically active substances in the extracellular compartment, provided that the patient is not in renal failure or does not have severe hyperglycemia.

Serum sodium has been proven to be an important prognostic factor in patients admitted to hospital. A recently published study confirmed that the relationship between sodium at admission and mortality in cardiac intensive care units and in hospital was a J-shaped curve. Patients with hypernatremia were those with the high risk of adverse outcomes, more severely burdened with comorbidities, and with increased adjusted mortality after discharge [14]. The authors of this study associated the results obtained with the possible negative impact of hypernatremia on left ventricular systolic performance, insulin sensitivity, or nerve and muscle function, because such correlations had been described in the literature, but also took into account the likelihood of the worsening prognosis of comorbidities that increase the possibility of elevated sodium levels. Underhydration could be one of the mechanisms that may be associated with poor prognosis in hospitalized elderly patients with hypernatremia. Studies that assessed the relationship between dehydration and serum sodium in older patients, published so far, most often used an arbitrarily determined sodium level higher than 145mmol/ L to recognize hypernatremia, and this approach could be sufficient for patients admitted to intensive care units. The results of the current study indicate that in geriatric patients, the possibility of underhydration should already be taken into account at lower (within ‘normal’ range) sodium concentrations. As such, perhaps, in the case of geriatric patients, the relationship between serum sodium levels higher than 140mmol/ L and short-term and long-term prognoses, should be evaluated.

The study has some limitations that should be considered. First of all, as mentioned above, the evaluated group of patients was not a randomly selected sample. Therefore, the results must be treated with caution, as predictive abilities of analyzed variables for the general population can differ. As this was a secondary analysis of data previously collected, some limitations resulted from that (some data was not available, as indicated in tables). As patients were classified based on calculated osmolarity, and not on plasma osmolality measurement results, the term "dehydrated" corresponds to subjects with impending underhydration in this study. On the other hand, the aim of the study was to find a simple way to catch cases at risk of underhydration, so as to be able to implement preventive measures early enough to prevent this condition from getting worse. Another limitation of the study was associated with validating serum sodium against a calculated value of which it forms a part; certainly, an analysis in relation to the measured osmolality would be more reliable, and it needs further research.

## 5. Conclusions

In summary, impending underhydration can concern about half of geriatric patients burdened with multimorbidity and frailty. It was shown that serum sodium could be treated as the main single test for the assessment of impending low-intake dehydration in older patients, but with the value equal to 140 mmol/L or higher as the best cut-off point suggesting its occurrence. It can be treated as a factor alerting carers to pay attention to proper fluid supply, and allowing for well-timed intervention, before underhydration occurs. It could be also considered as a first-step screening procedure in general practice, especially when limited resources restrict the possibility of more in-depth biochemical assessments. Perhaps we should consider changing the range of serum sodium norms for frail geriatric patients, whose resources for regulating water and electrolyte homeostasis are smaller, which may lead to dehydration more easily. However, this requires in-depth studies in which this relationship would be assessed based on measured osmolality, and studies from a longitudinal perspective.

## Figures and Tables

**Figure 1 nutrients-12-00496-f001:**
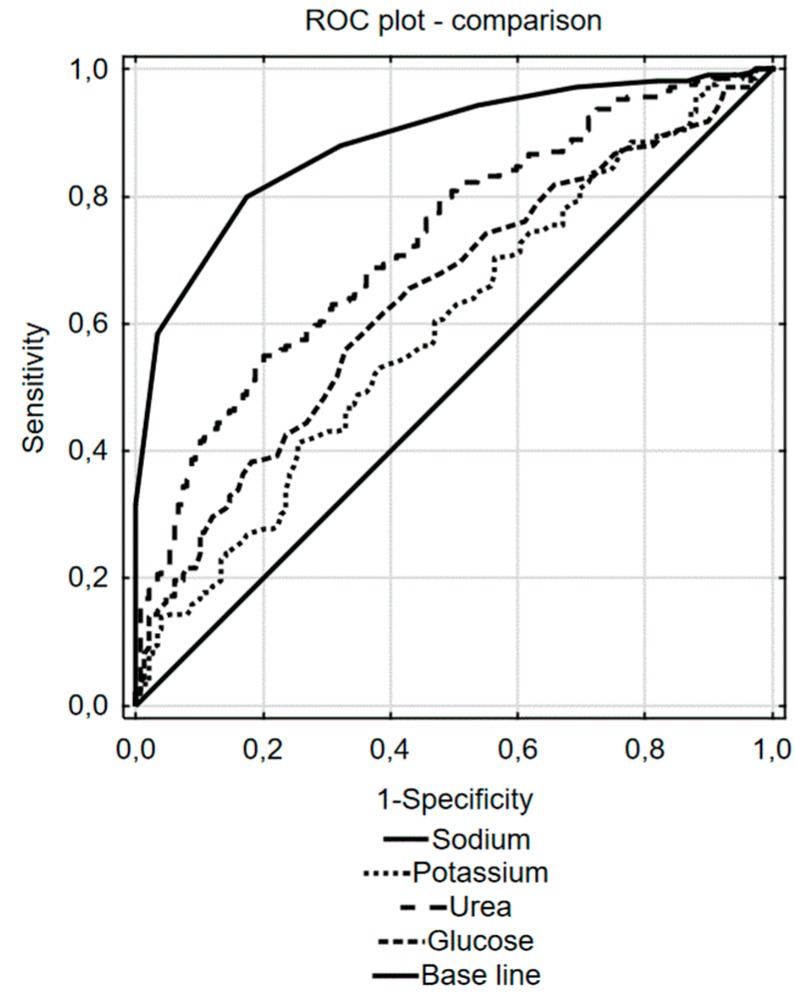
Receiver operating characteristics (ROC) curve analysis for the ability of sodium, urea, glucose, and potassium to predict impending underhydration.

**Table 1 nutrients-12-00496-t001:** Descriptive characteristics of participants at admission, stratified by hydration status.

	All *n* = 358 (100%)	Underhydrated *n* = 209 (58.4%)	Euhydrated *n* = 149 (41.6%)	*p* value^1^
Age, years, Me (IQR)	82 (78–86)	82 (78–86)	82 (77–86)	0.33
Gender				
Male, *n* (%)	84 (23.5)	48 (23.0)	36 (24.2)	0.80
Female, *n* (%)	274 (76.5)	161 (77.0)	113 (75.8)	
Living in long term care, *n* (%)	11 (3.1)	7 (3.3)	4 (2.7)	1.0
Living alone, *n* (%)	105 (30.3), *n* = 347	64 (31.7), *n* = 202	41 (28.3), *n* = 145	0.55
Hospitalization in the last year, *n* (%)	98 (27.6), *n* = 355	55 (26.6), *n* = 207	43 (29.1), *n* = 148	0.63
Number of chronic diseases, Me (IQR)	5.0 (3.0–6.0)	5.0 (4.0–6.0)	4.0 (3.0–6.0)	**0.001**
Multimorbidity, *n* (%)	210 (58.7)	134 (64.1)	76 (51.0)	**0.02**
Number of medications, Me (IQR)	7.0 (5.0–9.0), *n* = 350	7.0 (5.0–10.0), *n* = 206	7.0 (4.0–9.0), *n* = 144	**0.01**
Polypharmacy, *n* (%)	275 (78.6)	172 (83.5)	103 (71.5)	**0.008**
Barthel Index, Me (IQR)	90 (70–100), *n* = 356	90 (70–100), *n* = 208	90 (65–100), *n* = 148	0.48
IADL, Me (IQR)	7.0 (2.0–11.0), *n* = 352	7.0 (2.0–11.0), *n* = 206	7.0 (1.75–11.0), *n* = 146	0.55
AMTS, Me (IQR)	8.0 (6.0–9.0), *n* = 332	8.0 (6.0–9.0), *n* = 196	8.0 (6.0–9.0), *n* = 136	0.88
GDS, Me (IQR)	6.5 (3.0–10.0), *n* = 320	6.0 (3.0–10.0), *n* = 188	7.0 (4.0–10.0), *n* = 132	0.32
MNA-SF, Me (IQR)	11.0 (9.0–13.0), *n* = 349	12.0 (9.0–13.0), *n* = 203	11.0 (9.0–13.0), *n* = 146	0.26
CFS, Me (IQR)	5.0 (4.0–6.0)	5.0 (4.0–6.0)	5.0 (4.0–5.5)	0.69
Osmolarity, mMol/ L, Me (IQR)	292.9 (288.4–296.0)	295.4 (293.6–298.4)	287.7 (283.8–289.6)	**<0.001**
Serum sodium, mmol/L, Me (IQR)	140.0 (138.0–141.0)	141.0 (140.0–142.0)	138.0 (136.0–139.0)	**<0.001**
Na < 135 mmol/ L	24 (6.7)	4 (1.9)	20 (13.4)	**<0.001**
Serum potassium, mmol/L, M (SD)	4.43 (0.46)	4.50 (0.46)	4.33 (0.44)	**0.001**
Serum urea, mmol/ L, Me (IQR)	6.9 (5.5–8.8)	7.8 (6.2–9.9)	5.8 (4.9–7.3)	**<0.001**
Serum glucose, mmol/L, Me (IQR)	5.5 (5.0–6.3)	5.6 (5.2–6.7)	5.3 (4.9–5.8)	**<0.001**
Serum creatinine, mmol/L, Me (IQR)	86.2 (74.3–107.2), *n* = 354	91.05 (77.8–115.8), *n* = 207	78.7 (69.8–96.4), *n* = 147	**<0.001**
eGFR, ml/min/1.73m2, M (SD)	57.8 (17.1), *n* = 354	58.2 (16.8), *n* = 207	58.4 (16.7), *n* = 147	0.91

Underhydrated refers to subjects with serum calculated osmolarity > 295 mMol/L. ^1^—*p* values were determined by using a *t*-test for independent samples or Mann–Whitney test (continuous or interval variables) and χ2 test or Fisher exact test, as appropriate (categorical variables) to compare participants categorized according to hydration status. AMTS, Abbreviated Mental Test Score; CFS, 7-point Clinical Frailty Scale level; eGFR, glomerular filtration rate; GDS, 15 items Geriatric Depression Scale; IADL, instrumental activities of daily living; IQR, interquartile range; M, mean; Me, median; MNA-SF, Mini Nutritional Assessment-Short Form; n, number of cases; Na, sodium; SD, standard deviation. Bold signifies a statistically significant value.

**Table 2 nutrients-12-00496-t002:** The values of sodium, urea, glucose, and potassium for the prediction of impending underhydration, and their overall diagnostic effectiveness.

ROC Index	Sodium	Urea	Glucose	Potassium
AUC	0.88	0.730	0.64	0.60
95% CI of AUC	0.85–0.92	0.68–0.78	0.59–0.70	0.54–0.65
*p* value	**< 0.0001**	**< 0.0001**	**< 0.0001**	**0.002**
Youden Index J	0.625	0.349	0.231	0.156
Cut-off criterion	140 mmol/L	7.52	5.61	4.57
Sensitivity (%)	0.80	0.75	0.56	0.411
95% CI of sensitivity	0.74–0.86	0.68–0.80	0.48–0.62	0.34–0.48
Specificity	0.83	0.72	0.67	0.745
95% CI of specificity	0.75–0.88	0.65–0.79	0.59–0.75	0.67–0.81
Positive likelihood ratio	4.58	2.73	1.70	1.61
Negative likelihood ratio	0.24	0.35	0.66	0.79
**Difference between areas**				
Sodium		0.15 (0.10–0.21)	0.24 (0.18–0.30)	0.29 (0.23–0.35)
*p* value		**< 0.001**	**< 0.001**	**< 0.001**
Urea			0.09 (0.01–0.16)	0.14 (-0.06–0.21)
*p* value			**0.02**	**< 0.001**
Glucose				0.05 (-0.03–0.13)
*p* value				0.25

AUC, under the curve area; CI, confidence interval; ROC, receiver operator characteristic. Bold signifies a statistically significant value.

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
