# Peer review of "What Serum Sodium Concentration Is Suggestive for Underhydration in Geriatric Patients?"

_nutrients, 2020, doi:10.3390/nu12020496_

Round 1

Reviewer 1 Report

The goals of this study was to evaluate whether specific blood markers were useful in detecting dehydration in a cohort of geriatric patients and establish cutoff values for such markers. A secondary analysis of data collected from a previous study was used to achieve these goals. 

Based upon lines 125-126, the major finding from this work is that sodium was the better predictor for dehydration compared to urea, glucose and potassium. Based upon this finding, the major issue with this manuscript is that there isn't an explanation is the discussion as to why this finding is important or its relevance to human health. The title has sodium in it, but there isn't anything about sodium in the introduction or why sodium concentrations (or the other markers) are important to the detection of dehydration in geriatric patients. There is a lot of text about the diagnostic process and the Khajuria and Krahdan equation, but nothing regarding the public health relevance of this work. Why is this important?

Reviewer 2 Report

This is a generally well-written manuscript on a topic of interest to a wide variety of healthcare providers.  Analysis of the findings considering statistically significant differences the hydrated and dehydrated samples is required.  In addition, both introduction revisions and discussion of the results in light of the apparent purpose, recommendation of a lower threshold for serum sodium, are necessary.

Round 2

Reviewer 2 Report

Thank you for the detailed response to comments.  The manuscript is greatly improved.  Some minor grammar/punctuation needs to be corrected during proofing.  The only correction required at this time is line 254 - fail should be 'frail'.  
